# Disturbance of Immune Microenvironment in Androgenetic Alopecia through Spatial Transcriptomics

**DOI:** 10.3390/ijms25169031

**Published:** 2024-08-20

**Authors:** Sasin Charoensuksira, Supasit Tantiwong, Juthapa Pongklaokam, Sirashat Hanvivattanakul, Piyaporn Surinlert, Aungkana Krajarng, Wilai Thanasarnaksorn, Suradej Hongeng, Saranyoo Ponnikorn

**Affiliations:** 1Division of Dermatology, Chulabhorn International College of Medicine, Thammasat University, Pathum Thani 12120, Thailand; sasin.char@dome.tu.ac.th (S.C.); supasit.tantiwong@gmail.com (S.T.); juthapaa@gmail.com (J.P.); drwilai1330@gmail.com (W.T.); 2Chulabhorn International College of Medicine, Thammasat University, Pathum Thani 12120, Thailand; sirashat.han@dome.tu.ac.th (S.H.); piyaporn.latte@gmail.com (P.S.); krajarng@yahoo.com (A.K.); 3Research Unit in Synthesis and Applications of Graphene, Thammasat University, Pathum Thani 12120, Thailand; 4Division of Dermatology, Faculty of Medicine, Ramathibodi Hospital, Mahidol University, Bangkok 10400, Thailand; 5Division of Hematology and Oncology, Department of Pediatrics, Faculty of Medicine Ramathibodi Hospital, Mahidol University, Bangkok 10400, Thailand; suradej.hon@mahidol.ac.th; 6Thammasat University, Pattaya Campus, Bang Lamung 20150, Thailand

**Keywords:** androgenetic alopecia, spatial transcriptomics, bioinformatics, microinflammation, peri-infundibular immune infiltration

## Abstract

Androgenetic alopecia (AGA) is characterized by microinflammation and abnormal immune responses, particularly in the upper segment of hair follicles (HFs). However, the precise patterns of immune dysregulation remain unclear, partly due to limitations in current analysis techniques to preserve tissue architecture. The infundibulum, a major part of the upper segment of HFs, is associated with significant clusters of immune cells. In this study, we investigated immune cells around the infundibulum, referred to as peri-infundibular immune infiltration (PII). We employed spatial transcriptome profiling, a high-throughput analysis technology, to investigate the immunological disruptions within the PII region. Our comprehensive analysis included an evaluation of overall immune infiltrates, gene set enrichment analysis (GSEA), cellular deconvolution, differential expression analysis, over-representation analysis, protein-protein interaction (PPI) networks, and upstream regulator analysis to identify cell types and molecular dysregulation in immune cells. Our results demonstrated significant differences in immune signatures between the PII of AGA patients (PII-A) and the PII of control donors (PII-C). Specifically, PII-A exhibited an enrichment of CD4^+^ helper T cells, distinct immune response patterns, and a bias toward a T helper (Th) 2 response. Immunohistochemistry revealed disruptions in T cell subpopulations, with more CD4^+^ T cells displaying an elevated Th2 response and a reduced Th1-cytotoxic response compared to PII-C. These findings reveal the unique immune landscapes of PII-A and PII-C, suggesting potential for the development of innovative treatment approaches.

## 1. Introduction

Androgenetic alopecia (AGA) is the most common type of progressive hair loss in men and substantially affects an individual’s self-esteem and quality of life [1,2]. Currently, both oral finasteride and topical minoxidil solution or foam are considered FDA-approved standards for treating AGA in men [3]. However, the use of these treatment options is associated with unwanted side effects. Local irritation of the scalp and facial hypertrichosis can result from topical minoxidil, while finasteride is linked to sexual side effects, including fertility problems [4,5]. These adverse effects may inconvenience some patients and potentially impact treatment compliance, necessitating the exploration of new therapies.

The pathophysiological hallmarks of AGA are hair cycle dysregulation and hair follicle (HF) miniaturization [6,7], stemming from several factors including genetic predisposition, androgen hypersensitivity of dermal papilla (DP) cells, and environmental factors [8,9,10]. It has been long recognized that DP cells, along with hair follicle stem cells (HFSCs), are the key drivers of hair regeneration. Additionally, DP cells have been identified as pivotal in androgen-mediated changes associated with AGA [9]. Consequently, many studies have predominantly focused on investigating DP cells [11]. However, in recent years, the HF microenvironment has become a hot topic in hair biology. The local HF microenvironment contains various cell types. The interplay between these microenvironmental compositions and HF affects hair growth status. The HF immune system, which influences hair growth through a variety of milieus, is a crucial component of the HF microenvironment [12,13,14].

The HF is a complex mini-organ with distinct compartments crucial for hair growth and cycling [15]. The infundibulum is the uppermost segment, extending from the epidermis to the sebaceous duct opening [16]. Surrounding the infundibulum is the peri-infundibular area, which refers to the region immediately adjacent to the infundibulum [17,18,19]. In this study, we use the term “peri-infundibular immune infiltration (PII)” to denote the observable clusters of immune cells in close proximity to the infundibulum. In histological examinations of human HFs, the upper segment of the HF, particularly the infundibulum, was the most intensely immune-infiltrated segment even in the HF from healthy volunteers, indicating that it is an immunologically active site [20]. Because the largest cluster of immune cells is located in the vicinity of HFs, PII might be of great importance for HF–immune-cell interactions that regulate hair growth through short-range signals [21]. Although AGA is generally classified as non-inflammatory alopecia, histological and molecular evidence has shown chronic low-grade micro-inflammation with a disturbed immunological response in the upper segment of the HF, especially the area of PII [17,22,23]. It is an essential feature of the early stages of AGA that precedes subsequent perifollicular fibrosis in the latter stage of AGA [24], suggesting its crucial role in the progression of the balding process. However, to date, the molecular immunopathogenesis of AGA remains unclear.

Although immunohistochemistry (IHC) is the current standard method for evaluating infiltrating immune cells, it is difficult to investigate multiple molecular signatures of immune cells using this technique. In recent years, omics approaches have enabled high-throughput data-driven discovery of various inflammatory markers in AGA scalps [25,26,27]. However, despite their powerful capabilities, these methods require the dissociation of tissues and cells from their native context and average gene expression or protein abundance across whole samples, resulting in partial signal loss from relatively rare cells. This is especially true when they are in a specific location within a complex microenvironment, such as immune cells in the PII regions [28].

This study aimed to determine the intricate immune interactions within the peri-infundibular region of hair follicles in AGA patients compared to control donors. By leveraging the GeoMx digital spatial profiling technology, we quantified transcripts within the context of native tissue and cellular structures. This technology has been effectively applied to various types of formalin-fixed, paraffin-embedded (FFPE) tissues [29,30]. By integrating spatial transcriptomic data with traditional immunohistochemistry, we identified specific immune mechanisms contributing to AGA pathogenesis. Thus, this study provides novel insights into the immune-mediated mechanisms of AGA, potentially guiding the development of targeted therapies to modulate these immune responses in patients with AGA.

## 2. Results

### 2.1. Participants’ Demographic Data

The study enrolled seven patients diagnosed with androgenetic alopecia (AGA) and five control donors. The median age was consistent across both groups at 30 years (AGA group: n = 7; control group: n = 5), with no significant difference in age distribution observed (*p* = 0.873). All AGA patients were classified as having a Type III vertex as per the Hamilton–Norwood system and reported at least one first-degree relative with AGA. In contrast, control donors did not report any family history of AGA. There were no signs of scarring alopecia or dermatitis on the scalp among participants, confirming eligibility under this study’s inclusion criteria. Immunohistochemistry (IHC) analysis was conducted on tissue samples from all participants. For spatial transcriptome analysis, tissues were specifically collected from the first two participants in each group—two AGA patients, both aged 27, and the first two control donors, aged 28 and 30, respectively (Appendix A).

### 2.2. Assessing Overall Immune Cell Presence and Density in PII Regions with Immunostaining

To evaluate the presence and pattern of perifollicular immune infiltrates (Figure 1A), H&E staining and CD45 immunostaining were performed. Perifollicular immune cells were visually observed in varying quantities across the visible HFs. They were particularly concentrated around the upper segments of almost all the HFs from both patients with AGA and control donors (Figure 1B,C, and Appendix A). Next, the CD45^+^ area in the PII region of each HF observed in the sections was quantitatively evaluated. Overall, there was no significant difference in the CD45^+^ area between PII-A and PII-C (median 0.02332 vs. 0.02987, *p* = 0.438) (Figure 1C,D).

### 2.3. Selecting Regions of Interest (ROIs) and Validating Immune Cell Enrichment in PII Regions Using Spatial Profiling Techniques

Although increasing evidence suggests the potential role of a dysregulated immune response in AGA, the immunopathogenesis of AGA has not been fully understood [25,26,27]. We believe that specifically capturing the immune cells in the PII region, the most intense inflammatory perifollicular area, would reduce the noise from immune cells in other areas of the skin while also providing specific information regarding immune-HF interaction.

To determine the distinct molecular signatures of PII-A and PII-C, spatial transcriptome profiling was conducted on three PII regions from AGA patients (PII-A) and two from control donors (PII-C) using biopsied scalp tissue sections. CD45, a transmembrane glycoprotein extensively expressed on all hematopoietic cells except mature erythrocytes [31], was employed as a biomarker to identify immune cells within the complex tissue architecture [32,33]. Accordingly, immunofluorescence staining was utilized to delineate CD45^+^ immune cells, facilitating the selection of ROIs specifically within the infundibular region of the HFs (Figure 2A and Appendix A). We were able to capture three ROIs from three HFs of two patients with AGA, as well as two ROIs from two HFs of two control donors.

To further substantiate the enrichment of immune cells within PII-A and PII-C regions identified by the CD45 fluorescence imaging marker, we assessed the expression of protein tyrosine phosphatase receptor type C (*PTPRC*), quantified using GeoMx digital spatial profiling. This methodology, previously used in other studies for identifying immune cell-enriched areas [34], was adapted in our research. Specifically, we included additional ROIs from the bulk area of the upper HF and outer root sheath (ORS) to serve as internal controls for comparison with the PII regions. Our analysis demonstrated a significant increase in *PTPRC* expression within the PII regions compared to the other areas (Appendix A). This result not only confirmed the effectiveness of our digital spatial profiling-based approach but also verified that the digital spatial profiling technology accurately captured the target immune cells within the specified tissue regions.

### 2.4. Enhanced CD4^+^ T Cell-Mediated Immunity in PII-A Regions

Unsupervised clustering of the samples and a principal component analysis (PCA) were performed on the gene expression profiles to identify distinct clusters of the ROIs within the dataset. Clear segregation between the transcriptional profiles of the PII-As and PII-C groups was observed (Figure 2B,C), indicating a difference in gene expression between the two groups.

To gain insight into the altered biological processes in PII-A compared with PII-C, we conducted a gene set enrichment analysis (GSEA) on the expression dataset against the gene ontology biological process (GOBP) gene sets to identify the potential biological processes. Several biological processes that correlate with PII-A were significantly enhanced, including the ‘regulation of T cell-mediated immunity’, ‘CD4 positive alpha-beta T cell differentiation’, and ‘regulation of CD4 positive alpha-beta T cell differentiation’ (Figure 3A and Appendix A), suggesting that PII-A was associated with CD4^+^ T cell-mediated immunity.

Next, we utilized the spatial deconvolution technique to estimate the cellular proportion within the ROIs, which allows the mapping of cell-type composition and spatial organization within intact HF tissues [35]. The patterns of the estimated cellular proportions in the PII-A and PII-C groups were visually distinct, as demonstrated by the PCA plot (Figure 3B), suggesting different immune cell compositions between the two groups. The estimated cellular proportions of all ROIs are presented in Figure 3C and Appendix A. T cells were the most abundant, constituting up to half of the infiltrated cells in both groups. In accordance with the GSEA results, the T cell subtypes in PII-A showed apparent differences from those in PII-C. CD4 memory T cells (*p* = 0.047) were upregulated in PII-A, whereas CD8 memory T cells and NK cells were downregulated (*p* = 0.006 and 0.040, respectively) (Figure 3D and Appendix A).

### 2.5. PII-A Shows Enhanced Antigen-Presenting Cell-Related Molecules and a Potential Th2 Immune Bias

A differential expression analysis was performed to detect major transcriptional changes and related biological insights. The analysis revealed 180 DEGs (106 upregulated and 74 downregulated) (Figure 4A and Appendix A). To understand the altered pathways and biological processes in PII-A versus PII-C based on these DEGs, we performed an over-representation analysis (ORA) on the GOBP. DEGs were enriched for GOBP terms (false discovery rate (FDR) < 0.01) associated with immune response, including ‘immune system process’, ‘response to cytokine’, and ‘inflammatory response’ (Figure 4B and Appendix A), confirming that the regulation of immunological responses in PII-A deviated from that in PII-C.

Additionally, a PPI network was constructed using the STRING database and Cytoscape software version 3.9.1. To identify the genes that play a critical role in the PPI, the CytoHubba plugin for Cytoscape was used to identify the genes with high degree centrality. Hub genes were selected from the PPI network using the MCC algorithm of the CytoHubba plugin. According to the maximal clique centrality (MCC) scores, the top ten highest-scored genes, including *CD8A*, *CD4*, *PRF1*, *CD40*, *CCR5*, *CCL4L2*, *CXCR3*, *NKG7*, *HLA-DRA*, and *C1QC*, were selected as hub genes (Figure 4C and Appendix A). Among these hub genes, *CD4* and antigen presentation-associated genes, including *CD40* and *HLA-DRA*, were upregulated. In contrast, the genes associated with cellular immune response and cytotoxicity, including *CD8* [36], *PRF1* [37], and *NKG7* [38], and interestingly, the T helper 1-associated marker *CXCR3* [39], were downregulated.

To identify upstream regulators driving DEGs, an upstream regulator IPA was used. Interestingly, among the predicted upstream regulators with an overlap *p*-value < 0.05 and absolute z-score > 2, *TBX21* (*p*-value of overlap < 0.001, activation z-score = −2.204), a key transcription factor for Th1 commitment [40], was predicted to be an inhibited upstream regulator, whereas *IL13* (*p*-value of overlap = 0.004, activation z-score = 2.121), a signature cytokine of type 2 immune response [41], was predicted to be an activated upstream regulator (Figure 4D and Appendix A). These results imply that PII-A may skew toward the Th2 response instead of the Th1-cytotoxic response.

### 2.6. Density of CD4^+^ T Cells and Th2 Markers Predominates in PII-A

Next, the subsets of T cells, the most abundant and strikingly diverse cell type, were further investigated using IHC and digital image analysis (Figure 5A–C). First, CD4^+^ and CD8^+^ T cells were quantified using CD4^+^ and CD8^+^ areas. Overall, the results showed that CD4^+^ T cells and CD8^+^ T cells had a statistically significant positive correlation (correlation coefficient r = 0.73, *p* <0.05) and a moderate positive correlation with a trend towards significance (correlation coefficient r = 0.49, *p* = 0.078) in PII-A and PII-C, respectively (Figure 5H). Notably, PII-A showed a significantly higher CD4^+^ area (median 0.0248271 vs. 0.0101948, *p* = 0.022) but a lower CD8^+^ area (median 0.0024669 vs. 0.003962, *p* = 0.024) than PII-C (Figure 5D,E). Additionally, PII-A demonstrated a higher CD4^+^:CD8^+^ area ratio (0.0299) with a median CD4^+^:CD8^+^ area ratio of 9.130:1. Whereas PII-C demonstrated a median CD4^+^:CD8^+^ area ratio of 3.468:1 (Figure 5G), suggesting that CD4^+^ T cells might play a role in AGA pathogenesis. To investigate the possibility of FOXP3-expressing regulatory T cells [42], a subset of canonical CD4^+^ T cells, accounting for the observed increase in CD4^+^ T cells, we performed a comparative analysis of the FOXP3^+^ area between PII-A and PII-C. We demonstrated that the FOXP3^+^ area was not significantly different between the PII-A and PII-C groups (median 0.00035 vs. 0.00050, *p* = 0.107) (Figure 5F), indicating that non-regulatory CD4^+^ helper T cells likely account for the higher number of CD4^+^ T cells in PII-A. Furthermore, we conducted IHC staining for CD56, a marker of NK cells [43], and observed a decreased CD56^+^ area in PII-A compared to PII-C (median 0.0004 vs. 0.0015, *p* = 0.021) (Figure 5I,J), which is consistent with our earlier analysis.

Bioinformatics analyses revealed distinct regulatory patterns in Th1 and Th2 responses for PII-A and PII-C. Specifically, there is a possible skewing of immune cells in PII-A toward Th2 cells when compared to those in PII-C. To validate this observation, we employed IHC to characterize the protein expression of selected key transcription factors and cytokines associated with Th1 and Th2 responses. Although our transcriptome dataset did not show differential expression of *TBX21* and *IL13*, these genes were predicted to serve as significant upstream regulators. This indicates that, whether or not they exhibit differential expression, downstream genes in this region are altered. Therefore, we set out to confirm their protein expression using IHC staining for T-bet and IL-13. Additionally, we chose GATA3 [44] and IFN-γ [45] as supplementary markers for Th2 transcriptional regulators and Th1 cytokines, respectively.

While the areas of key cytokines, including the IFN-γ+ area (median 0.0094172 vs. 0.015787, *p* = 0.08906332) and the IL13^+^ area (median 0.014169 vs. 0.011868, *p* = 0.641), did not exhibit significant differences between the two groups, we observed notable disparities in key transcription factors. The T-bet+ area was significantly lower in PII-A (median 0.0002 vs. 0.0008, *p* = 0.048), while the GATA3^+^ area was significantly higher (median 0.0287 vs. 0.0156, *p* = 0.010) (Figure 6A–H). These findings support the hypothesis that PII-A displays Th2-biased immune responses. Furthermore, we conducted IHC staining for IL-17A, a cytokine associated with Th17 cells [46], which also play a critical role in skin inflammation. However, no statistically significant differences were observed in the IL-17A^+^ area between the PII-A and PII-C groups (median 0.000049 vs. 0.000176, *p* = 0.213) (Figure 6I,J).

## 3. Discussion

Spatial transcriptome profiling is a revolutionary technology that has transformed our exploration of disease mechanisms by combining histological features with gene expression analysis [47]. Recently, GeoMx digital spatial profiling has been applied to several types of FFPE tissues, revealing novel insights into the pathogenesis of various diseases [29,30]. Our study represents the first application of this advanced technology in FFPE scalp biopsies for investigative dermatology, particularly for AGA. Specifically, we utilized GeoMx digital spatial profiling and a series of bioinformatics analyses to elucidate the disturbed immunological response in AGA at PII regions, the area of the HF with the most notable immune infiltration, hoping to discover novel and specific immune signatures of these immunologically active areas. Compared to prior omics studies that focused on the entire HF [25,26], our spatial technique is more advantageous because it allows for analyses at the substructural level resolution of an individual HF and captures focal involvement within an intact, heterogeneous microenvironment, making it an efficient method for studying PII without tissue disruption and potentially valuable for any clinical or research specimen of the disease with specific pathological areas. It allowed us to screen for major molecular disruptions exhibited by various immune cell types located in the targeted areas. Here, we demonstrated that PII-A was associated with a higher expression of genes related to antigen presentation and CD4^+^ helper T cells. Intriguingly, we expand the insights into the inflammatory element of AGA by demonstrating that PII-A exhibits a higher level of Th2 response but a lower level of Th1-cytotoxic response compared with those of PII-C, indicating a specific immune response function in AGA.

Perifollicular infiltration was predominantly observed in the upper segment of the HF from both patients with AGA and control donors, consistent with previous reports [17,20], indicating the role of this area in immune surveillance. The concentration of immune infiltration around the upper part of the HF, particularly at the infundibulum, can be attributed to several factors. Firstly, the lower, transient cycling portion of the HF is immune privileged, while the upper permanent region is not [20]. Additionally, the presence of resident Langerhans cells in the upper part of the follicle, responsible for monitoring the follicular canal and potential pathogen invasion, may attract immune cells and contribute to the immune response in this region [48]. Moreover, the expression of chemokines varies across different regions of the HF, with the infundibulum expressing CCL20, a chemokine involved in immune cell recruitment. This attracts dendritic cells and other immune cells to the infundibulum region [49]. Lastly, given that the infundibulum is colonized by microbes, the presence of immune cells in the infundibulum region may also be influenced by the HFs microbiome [50]. These factors may collectively contribute to the observed immune infiltration around the infundibulum, highlighting the dynamic interplay between the HF structure, immune cells, and external factors in HF homeostasis.

AGA is traditionally viewed as a non-inflammatory type of hair loss. However, this characterization has been nuanced by observations of inflammatory infiltration, particularly in the balding scalp areas of AGA patients. While certain investigations have identified perifollicular immune infiltration in AGA cases without the inclusion of healthy control donors for comparison [51,52], only a few studies have utilized scalp samples from healthy volunteers as a control group [24,53,54,55].The findings from these studies have shown inconsistent results regarding the extent of inflammation when AGA samples are compared with those from control donors. For example, one study reported that moderate inflammatory infiltrates and fibrosis were observed in 36.8% of AGA patients, a significantly higher percentage than the 9.1% seen in controls, indicating a greater degree of inflammation in AGA [53]. Conversely, another study found perifollicular infiltration in a substantial majority of cases, present in 73% of AGA patients and 84% of control specimens, with no significant difference in the severity of inflammation between the two groups [55]. The variability in findings regarding inflammatory infiltration in AGA can be attributed to a range of factors, including the age of participants, the severity of their condition, and the different methodologies employed to measure and quantify inflammation. These discrepancies underscore the complexity of the inflammatory profile of AGA and highlight the need for standardized approaches in the study of this condition.

In this study, we focused on patients with early-stage AGA who all presented with a uniform disease severity classified as having a Type III vertex as indicated by the Hamilton–Norwood system. These patients were carefully matched with control donors of similar age to ensure comparability. By selecting a specific microscopic field at the infundibulum within vertical sections of the scalp and defining a consistent ROI, we effectively reduced the presence of dermal immune cells in our analysis. This methodological precision enabled us to focus exclusively on the immune cells located in the immediate vicinity of the HFs. Our results demonstrated that, although there was variability in the amount of immune infiltration, the overall level of immune infiltration within the PII regions showed no significant difference between the AGA group and the control group. These findings align with those of Valdebran et al. [55], suggesting that both AGA patients and controls exhibit comparable levels of immune infiltration, particularly within the specified area around the HF infundibulum. The similar immune cell density in both groups further emphasizes the subtle nature of microinflammation in AGA.

However, with the application of spatial transcriptome profiling, our study has provided a detailed view of the immune cell landscape within the PII regions, revealing differences in immune signatures between PII-A and PII-C. This analysis has set the stage for an in-depth examination of the specific immune cell types present in AGA.

A positive correlation between CD4^+^ and CD8^+^ T cells suggests that their frequencies are increased simultaneously within an individual HF. The CD4^+^:CD8^+^ area ratio of greater than 1 implies that CD4^+^ T cells are the predominant cell type in both PII-A and PII-C, which aligns with existing evidence that CD4^+^ T cells predominantly reside in the dermis compared to CD8^+^ T cells [56]. Nevertheless, bioinformatics and IHC analyses showed that PII-A was associated with a higher frequency of CD4^+^ T helper cells but a lower frequency of CD8^+^ T cells and NK cells compared to PII-C, indicating that an imbalance between T helper cells and cytotoxicity could possibly affect HF homeostasis, contributing to AGA. In line with our results, activated CD4^+^ T-cell infiltrates around the lower portions of the infundibulum have been found in specimens from the transitional area of patients with progressive alopecia [52]. Concordantly, female pattern hair loss, a condition usually considered to be the female counterpart of male AGA [57], showed histologically more intense CD4^+^ T cell infiltration around HFs in frontal samples [58]. This suggests that CD4^+^ T cells may play a role in the balding process.

Although various factors have been proposed to trigger microinflammation in AGA scalps [59], the exact contribution of these factors to the disturbance in immune homeostasis of the HF has not been established. Marked hyperplasia of Langerhans’ cells within the HF and connective tissue sheath at the lower infundibulum and an increase in antigen-presenting cell-related genes, including class II HLA, have been associated with AGA balding scalps, consistent with our findings [26,52,54]. The increase in antigen-presenting cells and class II HLA molecules indicates that in patients with AGA, robust extracellular antigens are presented to CD4^+^ T cells [60]. The infundibulum has been described as the location where HFs interact with external environments. The presence of resident Langerhans cells in the upper part of the HF is responsible for monitoring the follicular canal and potential pathogen invasion [48]. Additionally, given that the infundibulum is colonized by microbes, the presence of immune cells in the infundibulum region may also be influenced by the HFs’ microbiome [50]. Microbiome composition changes have been observed in AGA scalps, and this may contribute to altering the immune composition [61]. While the mechanistic insights are not fully established, indirect evidence suggests that environmental changes, including microbiome composition, may lead to extracellular antigen presentation to CD4^+^ T cells in AGA, further recruiting them to the infundibular area. Furthermore, despite unknown specific antigens, this aberrant antigen presentation may also subsequently influence CD4^+^ T cell differentiation and functional outcomes [62].

Our findings reveal an intriguing perspective in which PII-A displays a Th2-biased response with lower Th1 activity compared to PII-C. This observation is noteworthy, as mast cells, classically associated with Th2-mediated inflammation and typically recruited in type 2 immune responses [63], have been identified within the thickened connective tissue sheath in the balding scalps of individuals with AGA. The number of mast cells has shown associations with perifollicular fibrosis in the advanced stages of the disease [24,64]. Moreover, prostaglandin D2 (PGD2) and its receptor, a chemoattractant receptor-homologous molecule expressed on Th2 cells (CRTH2), are elevated in the balding scalp [26,27] and are known to exert inhibitory effects on hair growth [65]. While the characterization of CRTH2 in perifollicular immune cells has yet to be investigated [66], it is preferentially expressed on CD4^+^ effector Th2 cells and plays a pivotal role in mediating the activation of Th2 responses [67]. Consequently, our findings provide crucial evidence that type 2 inflammation, particularly a Th2-biased response, may have a significant role in AGA. Th2 responses are instrumental in regulating tissue repair following injury [68]. However, when type 2 cytokine-mediated repair processes become chronic or dysregulated, they can also contribute to the development of pathological fibrosis in various organ systems [68,69], potentially explaining the perifollicular fibrosis observed in AGA [17]. As such, the progressive follicular fibrosis in AGA, without evident destructive inflammation, may result from a subtle chronic inflammatory process with a Th2-biased response, thereby shifting the perifollicular immune environment towards a profibrotic state.

Notably, despite the absence of detectable differences in cytokine levels (IFN-γ and IL-13), the significant alterations in the key transcription factors (T-bet and GATA3) represent a vital finding. These transcription factors provide valuable insights into the underlying immune response in AGA. The undetectable differences in cytokine levels can be attributed to several factors. Cytokine secretion is often transient and subject to fluctuations over time, potentially resulting in variations in their tissue expression levels (11), which could contribute to the observed discrepancies between the actual levels of individual cytokines and their predicted activities. Additionally, the low abundance of cytokines within tissues, the secretory nature of cytokines, and the heterogeneous staining patterns of secreted cytokines can introduce complexities in quantitative measurements via IHC. It is also possible that the subtle immune disruptions in AGA may not be sufficiently sensitive for detecting differences in cytokine levels using the current techniques. However, the observed shifts in transcription factors favoring Th2 differentiation highlight a potential role for Th2 pathways in AGA.

This study utilized the paradigm-shifting spatial OMICs technology for the first time to investigate immune disruptions in the peri-infundibular regions of HFs in AGA, providing a novel perspective on the inflammatory elements of the disease. We observed key immunological disruptions characterized by an imbalance in immune cell types and response patterns. Specifically, there was a significant increase in CD4^+^ T cells and a pronounced shift towards a Th2-biased response. Concurrently, there was a decrease in cytotoxicity, marked by reduced numbers of cytotoxic CD8^+^ T cells and NK cells, which may be associated with the observed reduced Th1 response in AGA. These changes, potentially resulting from altered exposure to extracellular antigens, underscore a perturbed adaptive immunity rather than overall inflammation, contributing to the pathogenesis of AGA. The insights gained suggest that targeted treatment to normalize hair follicle-associated immune cells might be more promising than global immune suppression. Although limited by a small sample size, these findings lay a solid groundwork for further research. Ultimately, expanding research into the causal relationships between adaptive immunity and HF pathology will be crucial for developing potential immunomodulatory strategies to address the balding process effectively.

## 4. Materials and Methods

Figure 7 provides a comprehensive overview of the entire methodological process used in our study, facilitating a visual understanding of the sequential steps and techniques employed.

### 4.1. Patient Selection and Tissue Biopsy

This study was conducted in accordance with the Declaration of Helsinki and approved by the Human Research Ethics Committee of Thammasat University (MTU-EC-OO-6-085/64). Written, informed consent was obtained from all participants. This study included six clinically confirmed patients with AGA and four control donors (Appendix A). All patients with AGA presented a Type III vertex as indicated by the Hamilton–Norwood classification system and exhibited more than 20% miniaturization at the vertex, as demonstrated by dermoscope inspection. Healthy males with a normal-appearing scalp and hair miniaturization of less than 20% were classified as control donors. None of the participants showed signs of scarring alopecia, such as loss of follicular ostia or follicular hyperkeratosis, nor signs of dermatitis on the scalp. Additional criteria for exclusion included individuals who had undergone any prior hair treatments, including topical medications within the preceding 6 months; systemic medications (Finasteride within the previous year; Dutasteride within the last 18 months); non-medicated hair growth products or supplements in the prior 3 months, or any other hair growth interventions within the last 3 months; systemic illnesses; psychological disorders; or a history of hair transplantation.

A 4-mm punch biopsy was performed at the transitional zone of the balding vertex on each patient with AGA and the analogous area from each control donor. Considering the participant’s best interests, only a single biopsy was acquired from each participant; therefore, it was unavailable for both the horizontal and vertical sections. Despite the lack of quantitative and morphometric data for the HF, the vertical section shows the full dermal thickness in each section. It allows for precise identification of the inflammatory infiltrate and the relative location of immune infiltrates within defined HF structures [70]. Therefore, the scalp biopsies in our study were prepared for vertical sections. The tissues were immersed in 10% formalin and preserved in paraffin blocks for subsequent analysis.

### 4.2. Spatial Transcriptome Profiling

Tissue sections from two patients with AGA and two control donors were subjected to spatial transcriptome profiling (Appendix A), resulting in four tissue sections (patient with AGA = 2, control donor = 2 sections. Tissues were randomly placed on slides, mixing samples from PAs and CDs on both slides (two slides in total) to reduce the number of slides required for probe hybridization and inter-slide variability. All slides were processed according to the GeoMx digital spatial profiling RNA Slide Preparation manual, MAN-10101-01, and the GeoMx NGS Library Prep Readout, MAN-10133-03 (http://nanostring.com/support/support-documentation/ (accessed on 22 October 2021)). Briefly, The 5-μm sections of FFPE scalp tissues were deparaffinized and rehydrated, followed by—antigen retrieval—the WTA panel (18,676 targets) was hybridized to the slides overnight at 37 °C. Each target probe incorporates a unique photocleavable oligonucleotide barcode that can be quantified by next-generation sequencing (NGS). The next day, following stringency washes and blocking, fluorescently labeled antibodies to detect CD45 (Cell Signaling; clone D9M8I; catalog number 13917BF), and a DNA binding dye (Syto83) were applied to the slides for 1 h at room temperature.

Following washing, the prepared slides were immediately loaded into GeoMx digital spatial profiling and covered with an acquisition buffer. The PII region was defined as the dermal areas adjacent to the infundibulum, where clusters of CD45^+^ cells were clearly observed (Figure 2A). These areas were manually outlined as ROIs using the instrument’s web interface. The ROIs were expanded to include the immune cell clusters observed around the infundibulum. The size of each ROI was adjusted to ensure adequate capture of cells for transcriptome profiling (minimum of 200 cells). CD45^+^ cells inside ROIs were segmented for subsequent spatial transcriptome profiling. Oligonucleotide barcodes encoding the target genes were released from their target-specific portions using ultraviolet illumination and captured. Sequencing libraries were prepared according to the manufacturer’s instructions. Raw reads were collected and matched to target genes using NanoString’s custom oligonucleotide barcode-to-gene matching pipeline.

The final list of detectable genes was obtained by dropping genes with a limit of quantification (LOQ) of 1% coverage within replicates. The LOQ was calculated using the geometric mean and standard deviation of the negative probes in the dataset. Imputed raw read counts per gene were normalized to the third quartile. PCA was performed on the matrix expression values using R library PCAtools version 2.0.0. Unsupervised clustering was performed on the normalized transcriptomic profiles using the R library Pheatmap version 1.0.12.

### 4.3. Gene Set Enrichment Analysis

GSEA [71] was performed using GSEA version 4.2.2. Data from previously generated gene expression profiles perturbed in PII-A regions compared to PII-C regions were imported into GSEA software. In this study, we analyzed the expression profiles against GOBP gene sets available from the Molecular Signatures Database. GSEA was run using default parameters. The conventional cut-off value for statistical significance used in GSEA is a FDR of 0.25. In order to reduce the likelihood of false positive results, this study used FDR 0.1 as cut-off value for enriched gene sets. GOBP in PII-A and PII-C with significant enrichment results were demonstrated based on the normalized enrichment score (NES) and FDR. The ggplot2 R package was used to visualize the results, which showed dysfunctional pathways in PII-A compared with PII-C.

### 4.4. Cellular Deconvolution

Normalized transcriptomic profiles were appended to the SafeTME profile matrix. Deconvolution was performed using the SpatialDecon algorithm. Next, a principal component analysis was performed on the matrix of estimated cell proportions using the R library PCAtools version 2.0.0. To analyze the significant differential expression of different cell types, differences between PII-A and PII-C were analyzed using the unpaired *t*-test. A *p*-value < 0.05 was used as the cut-off.

### 4.5. Differential Expression and Functional Analysis

Differentially expressed genes (DEGs) between PII-A and PII-C were identified using an unpaired heteroscedastic *t*-test of log2-transformed normalized gene expression data. The significance threshold for differential gene expression was set at a *p*-value of 0.05 and an absolute Log_2_ fold change (|Log_2_FC|) of 1.

The DEGs reported previously were subjected to over-representation analysis (ORA) for GOBP terms using g: Profiler (https://biit.cs.ut.ee/gprofiler/gost (accessed on 30 April 2022)). Enriched terms with a Benjamini–Hochberg FDR < 0.01 were categorized as significant [72]. The ggplot2 R package was used to visualize the results.

The predicted functional and physical protein-protein interaction (PPI) networks were analyzed and visualized using the STRING app version 2.0.1 in Cytoscape version 3.9.1 [73]. The DEG lists were inputted, and interactions were assessed using a confidence cutoff of 0.4. In a co-expression network, the MCC algorithm was reported to be the most effective method for identifying hub nodes [74]. The MCC of each node was calculated using CytoHubba version 1.5.1, a plugin in Cytoscape [74]. In this study, the genes with the top 10 MCC values were considered hub genes. The parameters of CytoHubba used in this study were as follows: the top ten nodes ranked by MCC values, display options, first-stage nodes, shortest pathway, and expanded subnetworks.

The differential expression table with identifiers and corresponding Log_2_FC and *p*-value was uploaded into IPA (QIAGEN, Aarhus, Denmark, https://digitalinsights.qiagen.com/IPA (accessed on 30 April 2022)). The core analysis function, upstream regulator analysis, was used to interpret differentially expressed data. The predicted upstream regulators were identified using the threshold of *p*-value < 0.05 and absolute activation z-score (|z-score|) > 2.

### 4.6. Immunohistochemistry Assay and Digital Image Analysis

The analyses in this study included seven patients with AGA and five control donors. Both groups were similar in age, with a median age of 30 (n = 7) and 30 (n = 5), respectively (*p* = 0.873). Then, 5-μm sections were cut and stained with hematoxylin and eosin (H&E) and the following immunohistochemical markers: CD45, CD4, CD8, and forkhead box P3 (FOXP3) [regulatory T cells (Tregs)], T-box expressed in T cells (T-bet) [T helper (Th) 1 transcription factor], Interferon-gamma (IFN-γ) [Th 1 cytokine], Interleukin 13 (IL-13) [Th 2 cytokine], and GATA Binding Protein 3 (GATA3) [Th 2 transcription factor], CD56 [NK cells], IL-17A [Th 17 cytokine] (Appendix A).

The slides were digitized using a MoticEasyScan (Motic Digital Pathology) slide scanner. The images of immunohistochemically stained slides were processed in software for digital bioimage analysis, QuPath v.0.1.2 (Queen’s University, Belfast, Northern Ireland) [75], an open-source software for image analysis that has been used to quantify inflammatory cell density in skin [76] and other tissues [77]. Briefly, we identified HF epithelial compartments. As illustrated in Figure 2A, the PII region is defined as the dermal tissue surrounding the infundibulum of the HF, where immune cells are present. For the immunohistochemistry analysis, a more stringent approach was employed to delineate the ROIs compared to spatial transcriptome profiling. There was no limitation on the minimum number of cells required for IHC analysis. Therefore, the basement membranes of the HF epithelium at the infundibulum level were manually annotated, and these annotations were expanded by 100 μm to capture the corresponding 100-μm perifollicular dermal zone. Consequently, only the mesenchymal zone within a 100-μm distance from the basement membrane was considered as the ROI for IHC analysis. A representative microscopic image from each slide was used as a training image. Each cell staining of interest was manually outlined and set as ‘positive’ while representative remaining dermal areas were set as ‘negative’ to train pixel classifiers, as previously described [76]. The pixel classifier for each immunohistochemical marker was used to define positive and negative areas (mm^2^) within each ROI. The positive area was normalized to the total dermal area (positive area/dermal area).

Statistical analyses were performed using GraphPad Prism version 9 and RStudio. The Mann–Whitney U test with a *p*-value < 0.05 was used to determine the significant difference between PII-A and PII-C. Plot visualization was constructed using GraphPad Prism version 9 and the R package ggplot2. Correlation analysis and visualization were performed using the R package corrplot version 0.93.

## Figures and Tables

**Figure 1 ijms-25-09031-f001:**
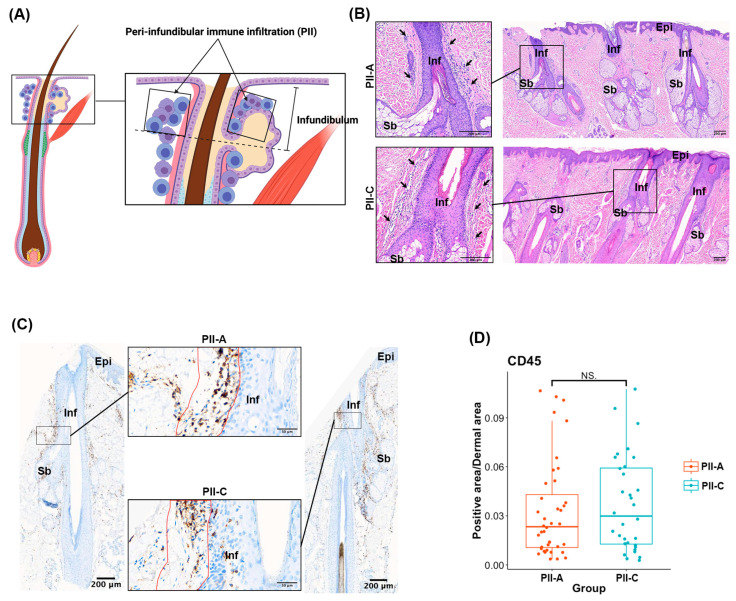
Evaluation of the peri-follicular and peri-infundibular immune infiltration. (**A**) Schematic illustration of a hair follicle highlighting the peri-infundibular immune infiltration (PII) region. (**B**) Representative scanned image of the H&E-stained sections (scale bar: 200 µm). The inset (scale bar: 200 µm) shows the infundibular regions of the HF. Black arrows indicate PII-A and PII-C. (**C**) Representative images of IHC staining for CD45 (scale bar: 200 µm). The inset (scale bar: 50 µm) shows the PII-A and PII-C. The red demarcation line indicates the boundary for the selected ROI for the QuPath analysis. (**D**) Box plot showing the median and IQR of the positively stained cell area/total dermal area of CD45^+^ area (PII-A = 42 fields, PII-C = 32 fields). The center line of the box represents the median value, the box represents the IQR, and the whiskers represent the range of data points within 1.5 IQRs of the median. We obtained 1–2 fields from each HF. Level of significance: NS = not significant. Epi, Epidermis; Inf, infundibulum; Sb, sebaceous gland; IQR, interquartile range.

**Figure 2 ijms-25-09031-f002:**
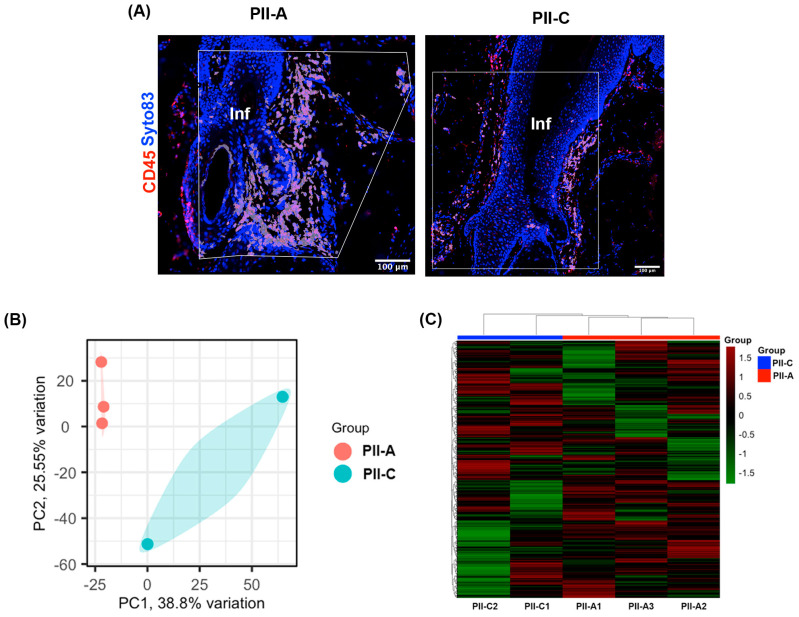
Region of interest (ROI) selection for spatial transcriptome profiling and evaluation of the expression patterns of each ROI. (**A**) Representative images of tissue sections stained with CD45 (red) and Syto83 (blue) (scale bars: 200 µm). The white polygonal lines indicate the boundaries for ROI selection. CD45^+^ areas within the ROIs were subjected to spatial transcriptome profiling. (**B**) PCA analysis of the transcriptional profiles of all ROIs. (**C**) Cluster heatmap of genes across all ROIs. The relative expression was scaled from red (high expression) to green (low expression). Each column represents an ROI (PII-A in red and PII-C in blue), and each row represents a gene. The figure presents 3 PII-A ROIs from 3 HFs from 2 patients with AGA and 2 PII-C ROIs from 2 HFs from 2 control donors.

**Figure 3 ijms-25-09031-f003:**
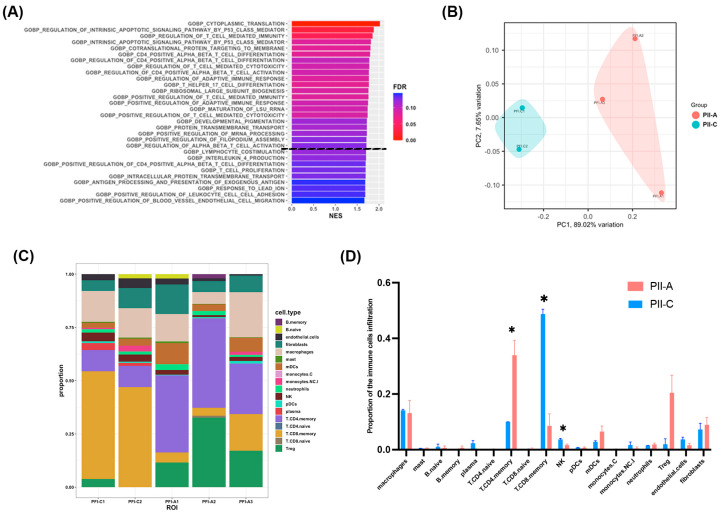
Gene set enrichment analysis (GSEA) and cellular deconvolution. (**A**) The bar chart represents the top significantly positively enriched gene sets defined by GSEA software version 4.2.2 using GOBP gene sets for genes in PII-A compared with PII-C. The *y*-axis represents the enriched GOBP gene sets, and the *x*-axis represents the NES of the enriched gene sets. The GOBP gene sets above the black dotted line are significant gene sets with an FDR < 0.1. The color of the bars represents the range of FDR, with lower FDR values displayed in red and higher FDR values shown in blue. (**B**) A PCA of the estimated relative proportions of cell types in a given ROI obtained by deconvolution of spatial transcriptome data using the SpatialDecon algorithm. (**C**) The stacked bar plots show the estimated relative proportions of each cell type in each ROI. (**D**) The bar graph shows the differential expression of different types of immune cells in PII-A and PII-C. The bars represent the mean ± SEM of the proportion of cells estimated in the ROIs. Level of significance: * *p* < 0.05.

**Figure 4 ijms-25-09031-f004:**
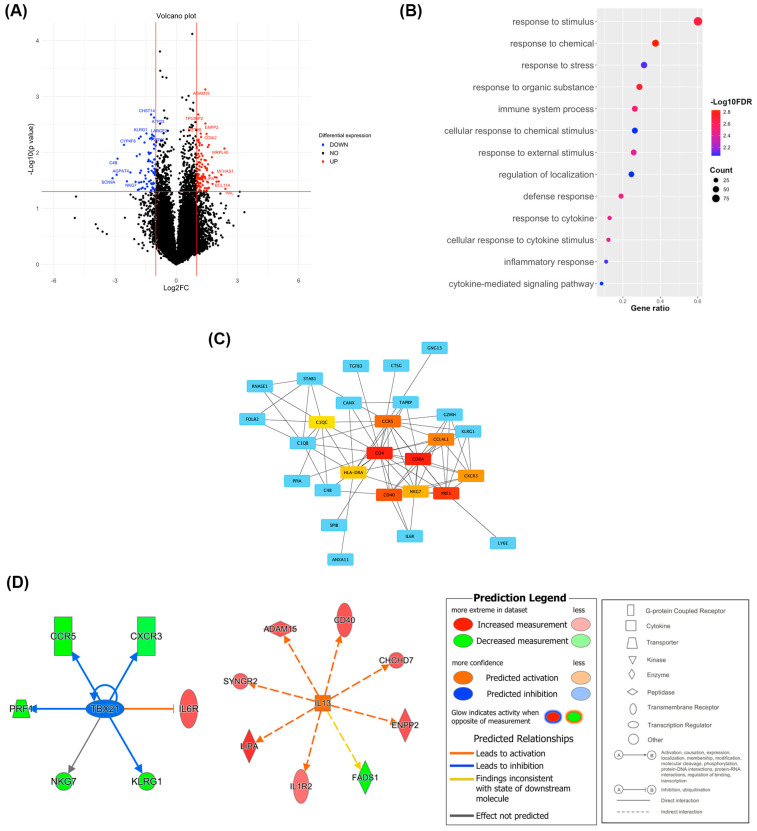
Bioinformatic screening for DEGs and their related biological processes, interactions, and regulators. (**A**) Volcano plot of DEGs identified between PII-A and PII-C. (**B**) Dot plots of GOBP terms that are significantly over-represented in the set of DEGs with FDR < 0.01. The *y*-axis represents the significant GOBP terms plotted in order of gene ratio. The gene ratio on the *x*-axis denotes the ratio between intersection and query size. The size of each dot or count refers to intersection size, i.e., the number of DEGs associated with the GOBP term, and the color of the dots represents the −log10 (FDR) obtained for the terms with the Fisher’s exact test. (**C**) The network highlights the top 10 hub genes ranked by the MCC algorithm and their neighbors using the CytoHubba Cytoscape plugin. The hub genes are in the center of the network (red-orange-yellow). The surrounding blue nodes are the neighboring molecules that relate to the hub genes in the center. The edge connecting two nodes represents the presence of any interaction between them. The connection degree level (degree of importance) of hubs is represented by a color scale ranging from red (highest degree) to yellow (lower degree). (**D**) The top two significant upstream regulators that have the highest degree of directional consistency (z-score), indicating consistent inhibition (blue) of *TBX21* and activation (orange) of *IL13*. Upstream regulators and their targets (surrounding molecules) are displayed as a network of interactions with their respective expression trends. Up-regulated and down-regulated genes are highlighted in red and green, respectively, and the color depth is correlated to the fold change. Orange, blue, yellow, and gray lines with arrows indicate activation, inhibition, inconsistent effect, and no prediction, respectively. Dash lines indicate indirect effects. Node shapes represent functional classes.

**Figure 5 ijms-25-09031-f005:**
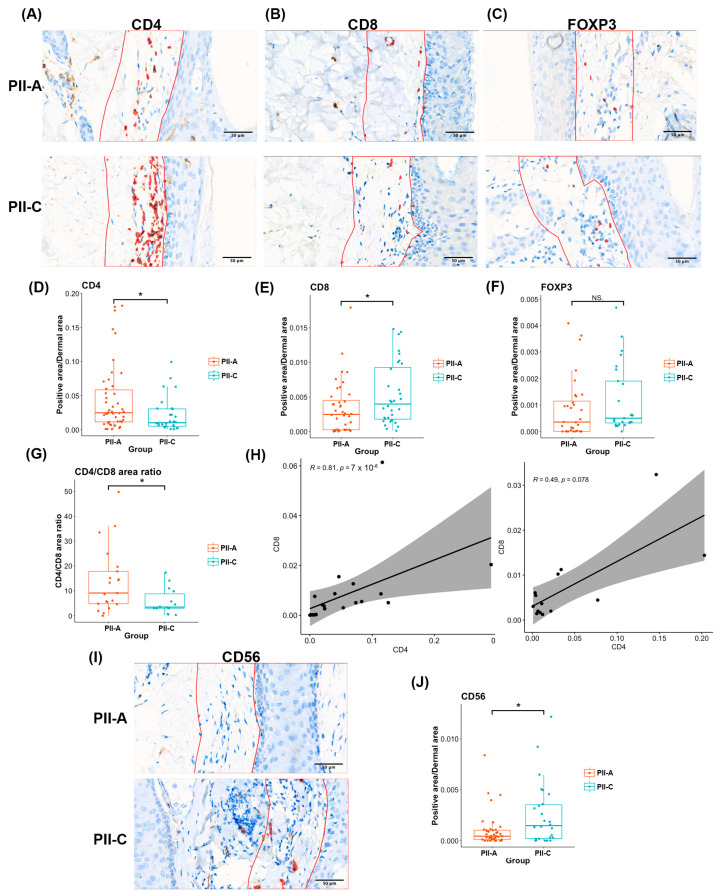
CD4, CD8, and FOXP3 expressions in PII-A and PII-C. (**A**–**C**) Examples of QuPath-analyzed CD4-stained sections (left), CD8-stainted sections (middle), and FOXP3-stained sections (right) with the coloring of positively stained immune cells (red). The red demarcation line indicates the borders of ROIs. (**D**–**F**) Box plots show the median and IQR of positively stained cell area/total dermal area of the CD4^+^ area (PII-A = 40 fields, PII-C = 27 fields): (**D**) CD8^+^ area (PII-A = 42 fields, PII-C = 35 fields); (**E**) and FOXP3^+^ area (PII-A = 30 fields, PII-C = 27 fields); and (**F**) The center line of the box represents the median value, the box represents the IQR, and the whiskers represent the range of data points within 1.5 IQRs of the median. (**G**) Box plot showing the median and IQR of the normalized CD4^+^:CD8^+^ area ratio (PII-A = 21 fields, PII-C = 14 fields). The center line of the box represents the median value, the box represents the IQR, and the whiskers represent the range of data points within 1.5 IQRs of the median. (**H**) Spearman’s correlation analysis between normalized CD4^+^ and CD8^+^ areas in PII-A (left) and PII-C (right). (**I**) Examples of QuPath-analyzed CD56-stained sections with the coloring of positively stained immune cells (red). The red demarcation line indicates the borders of ROIs. (**J**) Box plots showing the median and IQR of positively stained cell area/total dermal area of the CD56^+^ area (PII-A = 36 fields, PII-C = 31 fields). The center line of the box represents the median value, the box represents the IQR, and the whiskers represent the range of data points within 1.5 IQRs of the median. In this analysis, 1–2 fields were obtained from each HF. Level of significance: NS = not significant and * *p* < 0.05. IQR, interquartile range.

**Figure 6 ijms-25-09031-f006:**
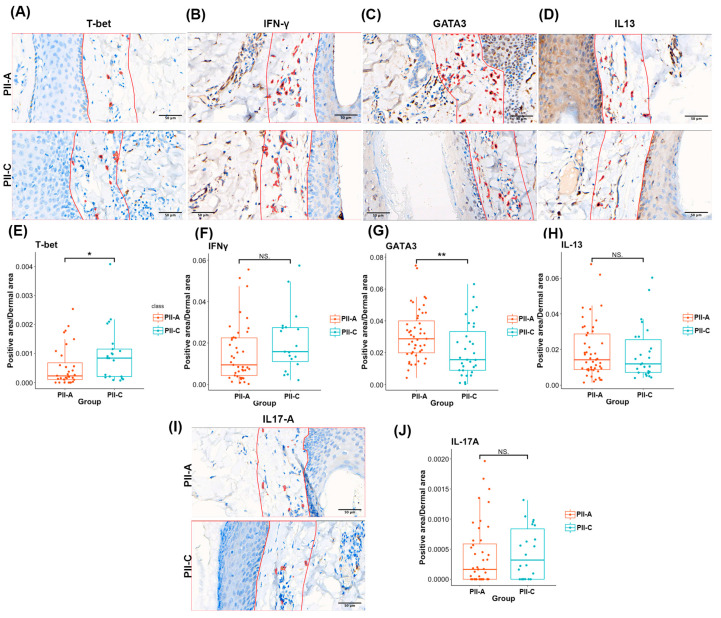
Th1 and Th2 marker expressions in PII-A and PII-C. (**A**–**D**) Examples of QuPath-analyzed T-bet-stained sections; (**A**) IFN-γ-stained sections; (**B**) GATA3-stained sections; (**C**) and IL13-stained sections; and (**D**) with the coloring of positively stained immune cells (red). The red demarcation line indicates the borders of ROIs. (**E**–**H**) Box plots showing the median and IQR of positively stained cell area/total dermal area of T-bet^+^ area (PII-A = 25 fields, PII-C = 30 fields): (**E**) IFN-γ^+^ area (PII-A = 41 fields, PII-C = 22 fields); (**F**) GATA3^+^ area (PII-A = 48 fields, PII-C = 34 fields); (**G**) IL13^+^ area (PII-A = 49 fields, PII-C = 29 fields); and (**H**) the center line of the box represents the median value, the box represents the IQR, and the whiskers represent the range of data points within 1.5 IQRs of the median. (**I**) Examples of QuPath-analyzed IL-17A-stained sections with coloring of positively stained immune cells (red). The red demarcation line indicates the borders of ROIs. (**J**) Box plots showing the median and IQR of positively stained cell area/total dermal area of IL-17A^+^ area (PII-A = 43 fields, PII-C = 29 fields). The center line of the box represents the median value, the box represents the IQR, and the whiskers represent the range of data points within 1.5 IQRs of the median. In this analysis, 1–2 fields were obtained from each HF. Level of significance: NS = not significant, * *p* < 0.05 and ** *p* < 0.01. IQR, interquartile range.

**Figure 7 ijms-25-09031-f007:**
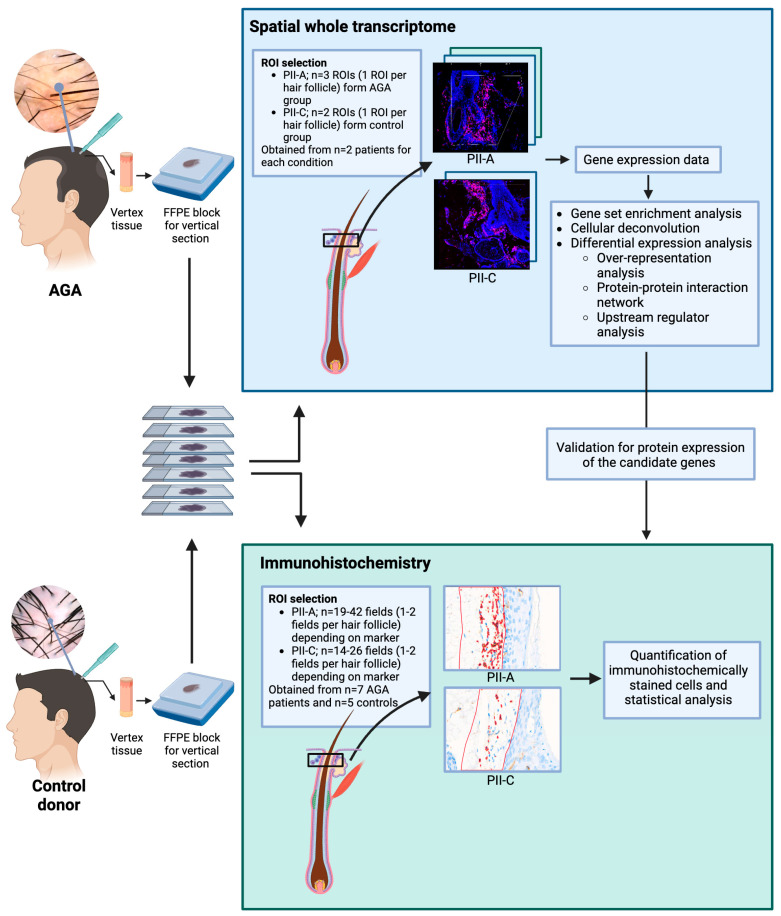
Workflow diagram of the methodology used in this study.

## Data Availability

The datasets generated and analyzed for this study can be found in NCBI Gene Expression Omnibus (GEO; http://www.ncbi.nlm.nih.gov/geo (accessed on 11 August 2023)) and are accessible through the GEO accession number GSE240637.

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
