# Peer review of "Disturbance of Immune Microenvironment in Androgenetic Alopecia through Spatial Transcriptomics"

_ijms, 2024, doi:10.3390/ijms25169031_

Round 1
Reviewer 1 Report
Comments and Suggestions for Authors
Interesting and innovative manuscript and study, there are some flaws that need to be addressed:
1. The last paragraph of the introduction section should be the aim of the study.
2. In the methodology or results section, it is necessary to write the demographic characteristics of the patients.
3. A limitation of the study is the small sample. The authors should have included a larger number of subjects.
4. The conclusion is too general, it should be written more specifically related to the results of the study.
5. authors should avoid excessive use of abbreviations, which make it difficult to follow the text.
6.it would be suitable to use abbreviations for groups uniformly, PA and CD or PII-A and PII-C.
Author Response
Reviewer 1
Comments and Suggestions for Authors
Interesting and innovative manuscript and study, there are some flaws that need to be addressed:
- The last paragraph of the introduction section should be the aim of the study.
Response:
Thank you for your insightful feedback on our manuscript. In response to your comment, we have revised the last paragraph of the introduction to more explicitly state the aims of the study. This revision helps to emphasize the methodological approach and the expected contributions of our research to understanding AGA's immune-mediated mechanisms, potentially guiding the development of targeted therapies. We appreciate your guidance and believe that these modifications have significantly improved the clarity of our study's objectives.
- In the methodology or results section, it is necessary to write the demographic characteristics of the patients.
Response:
We have addressed this by adding a detailed description of the participants' demographic data at the beginning of the results section. This addition ensures that the characteristics such as age, clinical classification of AGA, and family history are clearly outlined. Additionally, we have included detailed exclusion criteria within the 'Patient Selection and Tissue Biopsy' section of the methodology to enhance clarity and provide a comprehensive understanding of the participant selection process.
- A limitation of the study is the small sample. The authors should have included a larger number of subjects.
Response:
The study's limitations, particularly the small sample size, are recognized and were influenced by multiple factors. Budget constraints, set by our funding, restricted the number of samples and regions of interest (ROIs) we could analyze.
Additionally, acquiring tissue from AGA patients typically involves samples from more advanced stages of the disease, as these are most often obtained during hair transplant procedures and may not fully represent the early pathological changes we aimed to study. To better address this, we selected patients at Hamilton-Norwood stage III, representing a relatively early stage of AGA. However, this presented its own challenges, as obtaining biopsy tissue from patients at this stage is particularly difficult due to their general reluctance to consent to such invasive procedures. Moreover, obtaining control samples is equally challenging, as biopsies are not routinely performed on individuals without AGA.
Despite these challenges, the use of GeoMx Digital Spatial Profiling technology allowed us to gain significant insights, similar to other studies in the field that also operated with limited sample sizes [1]. We have clearly acknowledged these limitations in the discussion section of our manuscript. Our preliminary findings are promising, and we are committed to expanding upon this work in future studies by securing additional funding. This will enable us to increase the sample size and further validate and deepen our results.
- The conclusion is too general, it should be written more specifically related to the results of the study.
Response:
We acknowledge the need for a more specific conclusion that directly ties back to the detailed results of our study. We have revised the conclusion to more accurately reflect the specific immunological findings and their implications in AGA, particularly highlighting the distinct shifts in immune cell populations and their potential roles in the pathogenesis of AGA. We believe these revisions will provide a clearer, more direct synthesis of our results and their significance in the context of AGA research. These updates have been incorporated into the final paragraph of the discussion section, which serves as the conclusion in our revised manuscript.
- authors should avoid excessive use of abbreviations, which make it difficult to follow the text.
Response:
To enhance clarity and readability, we have eliminated abbreviations such as "digital spatial profiling," "patients with AGA," and "control donors" throughout the text. These terms have been replaced with their full forms. This modification ensures that the manuscript is accessible and straightforward for all readers, effectively eliminating any potential confusion related to the use of abbreviations.
6.it would be suitable to use abbreviations for groups uniformly, PA and CD or PII-A and PII-C.
Response:
For participant references: We have replaced all instances of 'PA' and 'CD' with their full forms—“patients with AGA” and “control donors,” respectively—when referring to participants as individuals. This change has been uniformly applied throughout the main text of the manuscript to enhance readability and ensure that the terms are comprehensible to all readers. However, the abbreviations 'PA' and 'CD' are retained in the legends of Supplementary Figures and Tables where space constraints and the need for quick reference justify their use. We have included clear descriptions of these abbreviations in the legends to prevent any ambiguity.
For regions of interest (ROI) within the tissue: We have retained the abbreviations PII-A and PII-C to refer specifically to regions of interest within tissue samples.
References
- Margaroli C, Benson P, Sharma NS, Madison MC, Robison SW, Arora N, et al. Spatial mapping of SARS-CoV-2 and H1N1 lung injury identifies differential transcriptional signatures. Cell Rep Med. 2021;2(4):100242.
Reviewer 2 Report
Comments and Suggestions for Authors
This study investigates the immune microenvironment in androgenetic alopecia (AGA) using spatial transcriptomics. The findings reveal significant immune disturbances, particularly a Th2-biased response in the peri-infundibular region, compared to controls. This nuanced understanding of immune cell interactions highlights the critical role of immune dysregulation in AGA pathogenesis. The results underscore the potential for developing targeted therapies that modulate specific immune responses, offering a promising avenue for new drug development. By providing detailed insights into the localized immune landscape, this research paves the way for innovative treatments aimed at normalizing hair follicle-associated immune cells.
The study provides compelling evidence of a Th2-biased immune response in the peri-infundibular region of hair follicles in androgenetic alopecia (AGA). Given these findings, advanced therapies targeting Th2 inflammation, such as dupilumab or tralokinumab, could potentially improve AGA in patients. These therapies, which are currently used for treating other Th2-mediated conditions like atopic dermatitis, could modulate the specific immune pathways implicated in AGA. Consequently, targeting Th2 inflammation may offer a novel therapeutic approach to mitigate hair follicle miniaturization and improve clinical outcomes in AGA patients receiving these treatments.
Author Response
Reviewer 2
Comments and Suggestions for Authors
This study investigates the immune microenvironment in androgenetic alopecia (AGA) using spatial transcriptomics. The findings reveal significant immune disturbances, particularly a Th2-biased response in the peri-infundibular region, compared to controls. This nuanced understanding of immune cell interactions highlights the critical role of immune dysregulation in AGA pathogenesis. The results underscore the potential for developing targeted therapies that modulate specific immune responses, offering a promising avenue for new drug development. By providing detailed insights into the localized immune landscape, this research paves the way for innovative treatments aimed at normalizing hair follicle-associated immune cells.
The study provides compelling evidence of a Th2-biased immune response in the peri-infundibular region of hair follicles in androgenetic alopecia (AGA). Given these findings, advanced therapies targeting Th2 inflammation, such as dupilumab or tralokinumab, could potentially improve AGA in patients. These therapies, which are currently used for treating other Th2-mediated conditions like atopic dermatitis, could modulate the specific immune pathways implicated in AGA. Consequently, targeting Th2 inflammation may offer a novel therapeutic approach to mitigate hair follicle miniaturization and improve clinical outcomes in AGA patients receiving these treatments.
Response:
Thank you for your insightful comments and suggestions regarding our manuscript. We appreciate your positive feedback and the constructive advice on potential therapeutic avenues. We wholeheartedly agree with your assessment. The suggestion to consider advanced therapies targeting Th2 inflammation, such as dupilumab or tralokinumab, is highly relevant and could indeed offer novel therapeutic approaches for AGA.
While advanced therapies targeting Th2 inflammation, such as dupilumab, have shown effects on hair growth in conditions like alopecia areata (AA) [1, 2], there is currently no direct clinical evidence supporting their efficacy in treating AGA. However, the theoretical basis for using Th2-targeted therapies in AGA is compelling, given the Th2-biased immune response observed in our study. It is important to note that AGA is a chronic condition, and treatment strategies may need to be adjusted accordingly, potentially requiring longer-term use. Additionally, our study highlights the altered immune response specifically in the peri-infundibular region, suggesting that treatment strategies targeting immune responses in this specific area may be more effective. Nevertheless, the manipulation of immune response should be fine-tuned and balanced, as it is like a double-edged sword [3]. A proper level and timing of immune modulation should be optimized, which requires further studies.
Ultimately, we believe that targeting Th2 inflammation represents a promising avenue for developing new treatments for AGA. The insights provided by our study, combined with the potential of existing Th2-targeted therapies, could pave the way for innovative treatment strategies aimed at restoring hair follicle health and function. Thank you once again for your valuable feedback. We are planning ongoing research to further characterize these findings and elucidate the causal relationships, ultimately aiming to translate these insights into clinical applications.
- Kulkarni M, Rohan CA, Travers JB, Serrao R. Long-Term Efficacy of Dupilumab in Alopecia Areata. Am J Case Rep. 2022;23:e936488.
- David E, Shokrian N, Del Duca E, Meariman M, Glickman J, Ghalili S, et al. Dupilumab induces hair regrowth in pediatric alopecia areata: a real-world, single-center observational study. Arch Dermatol Res. 2024;316(7):487.
- Pham C, Sung C, Juhasz M, Yuan J, Senna M, Khera P, et al. The Role of Antihistamines and Dupilumab in the Management of Alopecia Areata: A Systematic Review. J Drugs Dermatol. 2022;21(10):1070-83.
Round 2
Reviewer 1 Report
Comments and Suggestions for Authors
The authors have successfully responded to all objections raised.